# A Standardized Extract of Microalgae *Phaeodactylum tricornutum* (Mi136) Inhibit D-Gal Induced Cognitive Dysfunction in Mice

**DOI:** 10.3390/md22030099

**Published:** 2024-02-21

**Authors:** Jonathan Maury, Antoine Delbrut, Vanessa Villard, Rémi Pradelles

**Affiliations:** 1Research & Development Department, Microphyt, 713 Route de Mudaison, 34670 Baillargues, France; antoine.delbrut@microphyt.eu (A.D.); remi.pradelles@microphyt.eu (R.P.); 2Amylgen SAS, 2196 Boulevard de la Lironde, 34980 Montferrier-sur-Lez, France; vanessa.villard@amylgen.com

**Keywords:** microalgae, *Phaeodactylum tricornutum*, fucoxanthin, cognitive function, ageing, inflammation, oxidative stress, brain, memory, omega 3

## Abstract

The microalgae *Phaeodactylum tricornutum* (PT) is distinguished by its rich nutrient profile, characterized by well-documented neuroprotective activities, including fucoxanthin (FX), a major carotenoid and polyunsaturated omega-3 fatty acids (n-3 PUFA). The current study aims to evaluate the protective effects of a standardized extract of PT (Mi136) containing 2% FX on cognitive function, oxidative stress, and inflammation parameters in a mouse model of accelerated aging. Seventy-two (72) male mice were randomly assigned to the blank control group (BC), negative control group (NC), and four similar microalgae extract of PT groups (branded as BrainPhyt™) with different human equivalent doses to evaluate potential dose-response effects. From day 01 to day 51, mice in the BC group were injected with a 0.9% normal saline solution, while mice in all other groups were subcutaneously injected with D-galactose (D-Gal) at a dose of 150 mg/kg once per day, five days per week. Results indicated that, for the three higher microalgae extract of PT dose groups, spatial cognitive function, swim latency, and step-through latency impairments induced by chronic D-Gal intoxication were significantly and fully inhibited, with mean values similar to those in the BC group during each day of testing. Similar benefits were observed in biochemical analysis, specifically regarding brain and plasma levels of lipid peroxidation, TNF-α, and IL-6 markers. These data underscore the positive effects of a standardized extract of PT containing 2% FX on cognitive function parameters such as spatial working memory, long-term memory, and short-term memory through the regulation of oxidative stress and inflammation pathways.

## 1. Introduction

In developed countries, the phenomenon of accelerated population aging has been extensively documented over the past few decades, leading to significant shifts in public health policies [1]. A consequential aspect of this demographic transformation is the impending challenge of preserving cognitive function in the years to come, posing substantial implications for both societal and economic realms. Age-related cognitive decline is characterized as a non-pathological reduction in cognitive capabilities, encompassing aspects such as information processing speed, attention span, and, notably, working (or short-term) memory [2]. These alterations are rooted in normal and intricate physiological changes intricately linked to the aging process [3]. The onset of this cognitive decline remains a subject of contention, with studies reporting initiation as early as the third [4] or fourth decade [5], while others pinpoint the commencement in the sixth decade [6]. Regardless of the timing, cognitive impairment associated with aging is acknowledged as a primary symptom of degenerative brain disorders like Alzheimer’s Disease or other forms of dementia [7]. Some scholars posit that brain disorders may exert a more significant impact than cardiovascular diseases and cancers combined in the forthcoming years [8]. Consequently, preventing age-related cognitive decline is of paramount importance to mitigate the risk of developing debilitating brain diseases. Although the precise physiological mechanisms are not fully elucidated, inflammatory and oxidative stress pathways appear to play a pivotal role in the aging process [9,10]. As summarized in McGrattan et al.’s recent review (2019) [11], microglial macrophages in the brain become chronically active during aging, leading to sustained production of pro-inflammatory cytokines such as interleukin-6 (IL-6) and tumor necrosis factor-α (TNF-α). Elevated levels of these cytokines can trigger a cascade of neuroinflammatory processes, including neuronal death, reduced brain volume, or cortical thinning [12]. Additionally, there is well-documented evidence of increased oxidative stress in the aging brain [10,13]. The oxidative stress induced by an excessive surge in free radicals initiates chain oxidation reactions, causing cell damage and aging, consequently resulting in significant cognitive impairment, given the brain’s heightened susceptibility to free radical attacks. Given these potential mechanistic insights, compounds exhibiting antioxidant and/or anti-inflammatory properties may prove promising in preventing cognitive decline during aging, complementing existing pharmacological and behavioral approaches [8,11,14].

Within this domain, marine environments harbor an extensive biological diversity of microalgae, representing a vast reservoir of largely unexplored bioactive molecules, including carotenoids, pigments, fatty acids, peptides, and sterols, renowned for their antioxidant and anti-inflammatory properties. The cultivation of microalgae in photobioreactors, employing eco-friendly production processes, offers a sustainable alternative to address societal challenges, particularly in the preservation of biodiversity and landscapes [15,16]. Notably, the microalgae *Phaeodactylum tricornutum* (PT) stands out for its rich nutrient profile, featuring well-documented neuroprotective activities, such as fucoxanthin (FX), a major carotenoid, polyunsaturated omega-3 fatty acids (n-3 PUFA), including eicosapentaenoic acid (EPA) and docosahexaenoic acid (DHA), and phycoprostans [17,18,19,20,21,22]. Primarily, the preventive role of fucoxanthin in cognitive impairment is suggested by its robust singlet oxygen-quenching properties, mitigating oxidative stress and lipid peroxidation. Furthermore, this key carotenoid exhibits the ability to traverse the blood-brain barrier, modulating various signaling pathways (e.g., Nrf-ARE and Nrf2-autophagy) implicated in brain oxidative stress and inflammation associated with aging [9,20,21]. Secondly, concerning n-3 PUFA, a higher plasma EPA level has been correlated with reduced gray matter atrophy in several brain regions [23]. Moreover, EPA serves as an agonist of peroxisome proliferator-activated receptor alpha (PPARα), thereby activating or inhibiting various signaling pathways (e.g., decreasing neuroinflammation and oxidative stress, promoting autophagy and lipid metabolism), directly contributing to neuroprotection. As for DHA, its established benefits include improving cerebral blood flow and reducing inflammation [24]. Finally, phycoprostans exhibit a specific impact on neuronal mitochondria by diminishing membrane potential, resulting in reduced reactive oxygen species (ROS) production [22]. Thus, the different bioactivities of these diverse molecules give the microalgae extract of PT a substantial potential to enhance cognitive functions in aging. However, to our knowledge, no studies have yet validated this hypothesis. The present study aims precisely to assess the protective effects of a standardized extract of PT containing 2% FX on cognitive function, oxidative stress, and inflammation parameters in a mouse model of accelerated aging.

## 2. Results

### 2.1. Effect on Y-Maze Spontaneous Alternation Performance

#### 2.1.1. Y-Maze Spontaneous Alternation Performance

The spatial cognitive function was evaluated using the Y-maze test, and the results are shown in Figure 1. The spatial cognitive function of mice was impaired by D-GAL chronic intoxication, and the results showed that the mean spontaneous alternation behavior of the NC group (41.16%) decreased by approximately 26.39% compared to that of the BC group (67.56%). Microalgae extract of PT low dose group (BP120) significantly but partially improved (57.15%) spatial cognitive function impairment induced by D-Gal chronic intoxication (*p* < 0.001). For the three higher microalgae extract of PT dose groups, the spatial cognitive function impairment induced by D-Gal chronic intoxication was significantly and fully inhibited. In contrast, the Y-maze test results showed a similar number of total arm entries and indicated no differences in overall behavioral locomotor activity among all groups. Detailed individual values were presented in the Appendix A.

#### 2.1.2. Place Learning in the Morris Water Maze

To evaluate the long-term learning and memory function, a Morris water maze test was conducted, and the results are shown in Figure 2a,b. During the 5 days of testing sessions, the mean swim latency in the BC group gradually and significantly decreased from 82 s to 18 s. The results indicated significant long-term learning and memory impairment induced by D-GAL chronic intoxication. As shown in Figure 2a, the mean swim latency time in the NC group (From 90 s on day 1 to 52 s on day 5) is significantly lower than in the BC group, particularly for sessions during days 3 to 5. Microalgae extract of PT low dose group (BP120) significantly but partially improved (204.60 s) swim latency time increased by D-Gal chronic intoxication at days 3 to 5. For the three higher microalgae extract of PT doses groups, the swim latency impairment induced by D-Gal chronic intoxication was significantly and fully inhibited with a similar mean value than in the BC group during each day of tests. Regarding data from the probe test, the time spent in the target quadrant is significantly decreased in the NC group, while there are no differences in all Microalgae extract of PT groups compared to the BC group, as shown in Figure 2b. Detailed individual values were presented in Appendix A.

#### 2.1.3. Passive Avoidance Test

To evaluate the short-term learning and memory function, a passive avoidance test was conducted, and the results are shown in Figure 3a,b. The results indicated significant learning and memory impairment induced by D-GAL chronic intoxication. The mean NC group (124.70 s) showed a 51.42% decreased step-through latency time compared to that of the BC group (256.70 s). Regarding escape latency time, the mean value in the NC group (90.27 s) is significantly increased compared to the BC group (20.83 s). Microalgae extract of PT low dose group (BP120) significantly but partially improved (204.60 s) step-through latency time decreased by D-Gal chronic intoxication (*p* < 0.01). For the three higher microalgae extract of PT doses groups, the step-through latency impairment induced by D-Gal chronic intoxication was significantly and fully inhibited with a similar mean value than in the BC group. Regarding escape latency time, the benefits induced by each microalgae extract of PT doses evaluated are similar to step-through latency time, as shown in Figure 3b. Detailed individual values were presented in Appendix A.

### 2.2. Effect on Brain Lipid Peroxidation Level

The levels of brain (hippocampus) lipid peroxidation in mice, expressed in percentage of BC level, are shown in Figure 4. Compared with the BC group, the levels of brain lipid peroxidation in mice injected with D-gal were significantly increased (*p* < 0.0001). Microalgae extract of PT low dose group (BP120) very significantly but partially decreased the brain lipid peroxidation induced by D-Gal chronic intoxication (*p* < 0.0001). For the three higher microalgae extract of PT dose groups, the increase of brain lipid peroxidation levels produced by D-Gal chronic intoxication was significantly and fully inhibited.

### 2.3. Effect on Brain and Plasma Inflammatory Markers Level

The levels of brain (hippocampus) and plasma inflammatory cytokines, expressed as a percentage of BC levels group in mice, are shown in Figure 5. Compared with the BC group, the levels of brain TNF-α and IL-6 in mice injected with D-gal (NC group) were significantly increased (*p* < 0.0001). The two microalgae extract of PT lower doses groups (BP120 and BP235) very significantly but partially decrease the levels of brain TNF-α and IL-6 induced by D-Gal chronic intoxication, while in microalgae extract of PT higher doses groups (BP295 and BP370), it is fully decreased. As shown in Figure 5c,d, the benefits of microalgae extract of PT on brain (hippocampus) and plasma IL-6 levels are similar to those for TNF-α.

## 3. Discussion

Cognitive decline associated with aging is recognized as a significant public health concern, stemming from intricate physiological processes. Utilizing ingredients with antioxidant and/or anti-inflammatory functions emerges as a promising strategy for preventing cognitive decline in aging [8,11,14]. Microalgae have been identified as a rich source of bioactive molecules, encompassing carotenoids, pigments, fatty acids, peptides, and sterols, known for their neuroprotective effects through well-established signaling pathways. In this study, our aim was to investigate the effect of a standardized extract of *Phaeodactylum tricornutum* (PT) containing 2% fucoxanthin (FX) using a D-GAL mice model to assess its impact on cognitive function parameters through the regulation of oxidative stress and inflammation. Results revealed that the three higher doses of the microalgal extract of PT significantly and completely inhibited cognitive function impairments induced by chronic D-Gal intoxication. These inhibitory effects were comparable to the mean values observed in the control group (BC) during each daily test. Similar benefits were observed in biochemical analyses, demonstrating reductions in brain and plasma levels of lipid peroxidation, TNF-α, and IL-6 markers. These findings underscore the positive impact of the standardized extract of PT containing 2% FX on cognitive function parameters, including spatial working memory and short-term memory, achieved through the regulation of oxidative stress and inflammation pathways.

D-Galactose (D-Gal) mimics various behavioral and molecular features of brain aging in rodent models and is commonly employed to investigate the mechanisms of brain aging and anti-aging therapeutics in animal studies. As a physiological nutrient and reducing sugar, D-Gal reacts with the free amines of amino acids in proteins, leading to the formation of advanced glycation end products through nonenzymatic glycation. Excessive D-Galactose supply can contribute to the generation of reactive oxygen species (ROS) through oxidative metabolism and glycation end products. Rodents chronically injected with D-Gal for 6–10 weeks have been reported to exhibit a progressive decline in learning and memory abilities and increased free radical production in the brain [25,26]. Wu et al. (2008) demonstrated that chronic D-Gal supplementation induced a significant increase in latency to the platform and a decrease in swim speed in mice models during the Morris Water Maze test. Moreover, Wei et al. (2017) reported that cognitive and behavioral parameters were impaired in a dose-dependent manner following chronic ingestion of D-Gal, justifying the dose of 150 mg/kg once per day, 5 days per week used in the present study. In our study, we observed significant cognitive impairments and elevated oxidative stress and inflammation markers in mice subjected to chronic D-Gal ingestion. The cognitive decline was particularly pronounced, with decreases ranging from 40% to 50% in various cognitive function tests compared to the placebo group. These alterations were paralleled by a notable increase of 200–300% in mean values of brain levels of lipid peroxidation, TNF-α, and IL-6 compared to the BC group. These results affirm the validity of the D-Gal mice model employed in our study and reinforce the positive benefits observed in all four experimental study groups. Indeed, the results showed that the three higher doses of the microalgae extract of PT fully inhibited impairments induced by chronic D-Gal intoxication in three major cognitive function tests. These results are comparable to the benefits induced by several interesting ingredients such as astaxanthin, spearmint, DHA, Ginkgo biloba, fucoidan, or Spirulina in different in-vivo cognitive-dysfunction models [27,28,29,30,31,32,33,34]. Ren et al. (2022) aimed to investigate the effects of different doses of a dietary supplement enriched with micronutrients, phosphatidylserine, and DHA on cognitive performance using a D-galactose (D-gal) induced aging rat model. Interestingly, the dietary supplement intervention significantly improved cognitive performance in the Morris water maze test with a dose-response kinetics similar to that in the present study. Similar dose-response benefits were shown following 3 weeks of supplementation with fucoidan, a sulfated fucose-containing polysaccharide extracted from brown algae, in a D-Gal mice model [32].

In addition to the observed improvements in behavioral cognitive function parameters, our study demonstrates that the microalgae extract of *Phaeodactylum tricornutum* (PT) regulates neuroinflammation and oxidative stress following D-GAL induction in mice, exhibiting a dose-response effect. Neurons, being metabolically robust and highly active consumers of oxygen, serve as major producers of reactive oxygen species (ROS) in their normal function [35,36]. A primary source of ROS is the mitochondrial respiratory complex, and due to its proximity to the site of production, mitochondrial DNA is particularly susceptible to oxidative damage (e.g., lipid peroxidation), leading to membrane fluidity [37]. Natural antioxidant processes in the nervous system are impaired during aging, mainly due to a downregulation of the Nrf2 pathway activity, resulting in redox potential imbalance. This imbalance leads to long-lasting accumulated cell damage (e.g., lipids, proteins, DNA) and has been identified as a major contributor to aging-related cognitive function decline [35,36,37]. Our study revealed a significant increase in lipid peroxidation levels in the mice hippocampus following D-GAL induction. The hippocampus, along with the frontal lobes, is a major brain region affected by oxidative stress with aging, directly contributing to alterations in learning, spatial working, and episodic memory [38,39]. Thus, the regulation and decrease of hippocampus lipid peroxidation levels observed following supplementation with the microalgae extract of PT may explain the benefits highlighted in cognitive function tests. Similar to mitochondrial dysfunction, cell senescence, or the general weakening of immune functions, the accumulation of oxidative stress is a major source of the upregulation of the production of pro-inflammatory cytokines. These, in turn, activate inducible nitric oxide synthase (iNOS) and NADPH oxidase (NOX) [40,41]. During aging, microglial macrophages, the resident innate immune cells of the central nervous system, become chronically active, inducing sustained production of pro-inflammatory cytokines such as interleukin-6 (IL-6), tumor necrosis factor-α (TNF-α), and proteins like C-reactive protein (CRP), well-recognized contributors to cognitive function decline [11]. The increase in these cytokines, initiated and regulated by the activation of the redox-sensitive transcription factor NF-kB, can induce a cycle of neuroinflammatory processes, including neuronal death, reduced brain volume, or cortical thinning [12]. Interestingly, an increase in peripheral inflammatory markers may activate neuroinflammation via neuronal and hormonal pathways [42,43]. As demonstrated in the present study, the microalgae extract of PT significantly decreased the levels of IL-6 and TNF-α at both systemic and hippocampus levels.

The microalgae extract of *Phaeodactylum tricornutum* (PT) contains several intriguing bioactive compounds, particularly fucoxanthin and omega-3 (Table 1), which may account for the observed benefits of oxidative stress and inflammation [20,35,44]. Firstly, fucoxanthin is a major carotenoid that exhibits antioxidant and anti-inflammatory activity, and its neuroprotective role has been highlighted in various in-vitro and in-vivo models [44,45,46]. Upon ingestion, fucoxanthin is rapidly metabolized by digestive enzymes to fucoxanthinol before absorption in the intestines, subsequently converting to amarouciaxanthin A in the liver. Apart from its ability to bypass the blood-brain barrier and strong singlet oxygen-quenching properties, which prevent oxidative stress by inhibiting H2O2-induced neurotoxicity (e.g., brain/systemic lipid peroxidation), fucoxanthin may prevent oxidative stress-induced neuronal death through the activation of the PI3 K/Akt/Nrf2 pathway. It can induce autophagy via the activation of BECLIN-1 and inhibit neuronal apoptosis through the inhibition of caspase-3. This process involves the translocation of the Nrf2 protein from the cytosol to the nucleus, facilitating the detachment of the keap1 protein. Nrf2 induces the expression of genes that activate the neuron’s autophagy mechanism, eliminating intracellular waste caused by oxidative stress. Nrf2 also induces the expression of antioxidant enzymes such as HO-1 and NQO-1 by binding to the ARE in the promoter of antioxidant genes, preventing the accumulation of waste that amplifies oxidative stress [45]. Furthermore, fucoxanthin has been shown to inhibit the production of pro-inflammatory cytokines, including TNF-α, IL-6, and IL-1β, by inhibiting NF-κB and MAPK pathways in microglia, the major immune cells in the brain [44,46,47]. Secondly, eicosapentaenoic acid (EPA) and docosahexaenoic acid (DHA) are long-chain polyunsaturated fatty acids (LC-PUFAs) from the omega-3 family, essential for brain functions, given that the human brain comprises approximately 50–60% lipids [48]. EPA, in particular, acts as an agonist of the PPAR-α receptor (peroxisome proliferator-activated receptor alpha), allowing its activation and regulation of different signaling pathways. PPAR-α not only inhibits neuroinflammation and oxidative stress but also optimizes neurotransmission processes by altering membrane fluidity and increasing neurotransmitter release. LC-PUFAs also alleviate brain apoptosis through various mechanisms, such as reducing responses to reactive oxygen species, exerting an antiapoptotic action, or up-regulating the expression of antiapoptotic proteins and down-regulating the expression of apoptotic proteins, thus slowing down the apoptosis reaction [48].

Overall, these data are promising, advocating for the promotion of the microalgae extract of *Phaeodactylum tricornutum* (PT) as a natural and viable alternative in the dietary supplement arena for improving cognitive function in humans. Notably, Leonard et al. (2023) demonstrated that both acute and 30-day ingestion of two different doses of the same microalgae extract of PT tested in the present study, combined with guarana in equal quantities across the experimental groups, improved reaction times, reasoning, learning, executive control, attention shifting (cognitive flexibility), and impulsiveness in e-gamers [49]. Our research team is currently conducting two clinical studies to evaluate the efficacy of PT microalgae on the cognitive function of healthy elderly individuals (NCT04832412 and NCT05759910). While additional pre-clinical studies are necessary to better comprehend and refine the underlying mechanisms of action, it would be intriguing to investigate the impact of this microalgae extract on the gut-brain axis. Over the past several years, a growing body of literature has underscored the bidirectional relationship between the gut and the brain [50,51]. Specifically, disruptions in gut homeostasis, influenced by various factors such as diet, infections, aging, or antibiotic exposure, can lead to compromised intestinal barrier integrity through tight junction impairment and apoptosis of gut epithelial cells [51]. Subsequently, chronic systemic inflammation and oxidative stress play a pivotal role in the brain by regulating the activity of multiple metabolites and neurotransmitters associated with specific signaling pathways [50,51]. As outlined in the review by Fakhri et al. (2021), marine-derived natural ingredients extracted from various sources, including microalgae, present a promising approach to positively modulate intestinal disorders, inflammatory mediators, apoptosis, and oxidative stress, thereby contributing to neuroprotective properties.

## 4. Materials and Methods

### 4.1. Animals, Diet and Treatment Groups

Seventy-two (72) male Swiss mice, weighing 30–35 g, from JANVIER company (Saint Berthevin, France), were kept for housing, and experiments took place in Amylgen (Direction Régionale de l’Alimentation, de l’Agriculture et de la Forêt du Languedoc-Roussillon, agreement #A 34-169-002 from 2 May 2014). Animals were housed in groups with access to food and water ad libitum, except during behavioral experiments. They were kept in a temperature and humidity-controlled animal facility on a 12 h/12 h light/dark cycle (lights off at 07:00 pm). Mice were numbered by marking their tail with permanent markers. All animal procedures were conducted in strict adherence to the European Union directive of 22 September 2010 (2010/63/UE). The experiment design (protocol #02441) was approved by the Mediterranée Ethics Committee in January 2015.

As shown in Figure 6, mice were randomly divided into blank control group (BC), negative control group (NC), and 4 microalgae extract of PT (branded name: BrainPhyt^TM^) groups with different human equivalent doses evaluated (BP120, dose of 120 mg/day/70 kg; BP235, dose of 235 mg/day/70 kg; BP295, dose of 295 mg/day/70 kg; BP370, dose of 370 mg/day/70 kg) with 12 mice in each group. Microalgae extract of PT supplementation has been administered by incorporation into food pellets during the study duration (between day −28 and day 51, a total of 79 days) by Safe company (Augy, France) and kept at 4 °C. BC and NC groups received normal food pellets. Between day 1 and day 51, mice in the BC group were injected with 0.9% normal saline, while mice in all other groups were subcutaneously injected with D-galactose (D-Gal) at a dose of 150 mg/kg once per day, 5 days per week. The health of animals was checked daily, and the general aspects of animals, activity, and weight were observed daily. Acute or delayed mortality was checked daily.

### 4.2. Microalgae Extract of PT Production and Characterization

The microalgae extract of the PT production process was based on microalgae cultures in industrial photobioreactors, and all steps were realized in Microphyt (Baillargues, FRANCE). Microphyt has developed and operates its own proprietary production technology called CAMARGUE^®^, which allows controlled microalgae biomass production. CAMARGUE^®^ technology consists of a 1200 m serpentine glass piping circuit, with a 76 mm inside diameter and 4.5 mm wall thickness, folded horizontally in 24 straight lines with a vertical height of 3 m, 0.3 m wide, realizing a 50 m long tubular fence. The global description of the product manufacturer is presented in the Figure 7, knowing that the first step consisted of biomass production from the microalgae *Phaeodactylum tricornutum*:

The crude ethanolic extract is standardized with regard to active content by dilution with Medium Chain Triglycerides (MCT) and stabilized with antioxidants. Thus, the final product, called microalgae extract of PT, is firstly standardized to contain 2% of FX. The global composition of microalgae extract of PT is presented in Table 1.

The detailed composition of microalgae extract of PT regarding total chlorophylls, carotenoids, and fatty acids contents is presented in Table 2. Total chlorophylls and carotenoids were determined by a UV-Vis spectrophotometer, while omega-3 fatty acids were quantified by gas chromatography.

### 4.3. Y-Maze Spontaneous Alternation Performance

On day 43, all animals were tested for spontaneous alternation performance in the Y-maze, an index of spatial working memory. The Y-maze was designed according to Hiramatsu et al. (1999) and is made of grey polyvinylchloride [52]. Each arm is 40 cm long, 13 cm high, 3 cm wide at the bottom, 10 cm wide at the top, and converging at an equal angle. Each mouse was placed at the end of one arm and allowed to move freely through the maze during an 8-min session. The series of arm entries, including possible returns into the same arm, were checked visually. An alternation was defined as entries into all three arms on consecutive occasions. The number of maximum alternations is, therefore, the total number of arm entries minus two, and the percentage of alternation was calculated as (actual alternations/maximum alternations) × 100. Parameters included the percentage of alternation (memory index) and total number of arm entries (exploration index). Animals showing extreme behavior (Alternation percentage < 20% or >90% or number of arm entries < 10) were discarded from the calculation. No animal was discarded accordingly.

### 4.4. Place Learning in the Morris Water Maze

From day 44 up to day 49, all animals performed the Morris Water Maze to test spatial working memory [53,54,55]. The water maze consists of a circular pool (diameter 140 cm, height 40 cm) filled with water at a temperature of 23 ± 1 °C placed in a room with controlled light intensity, external cues (sink, contrasted posters, shelves), and water opacity rigorously reproduced. A transparent Plexiglas non-slippery platform (diameter 10 cm) is immersed under the water surface during acquisition. Training consisted of 3 swims per day for 5 days, performed between days 44 and 48, with 20 min inter-trial time. Animals were tested by a batch of 12 individuals. Start positions, set at each limit between quadrants, were randomly selected each day, and each animal was allowed a 90 s swim in order to find the platform, and it was left on it for 20 s. The median latency, expressed as mean ± S.E.M., was calculated for each training day. Working memory was specifically assayed by changing the platform location every day. A probe test (PT) was performed 24 h after the last swim, on day 49. The platform was removed, and each animal was allowed a free 60-s swim. The start position for each mouse corresponds to one of two positions remote from the platform location in counterbalanced order. The platform quadrant was termed the training (T) quadrant, and others opposite (O), adjacent right (R), and adjacent left (L) during retention. The time spent in each quadrant was determined. The results were expressed as time spent in the target quadrant (T) compared to the mean of the three other quadrants (O).

### 4.5. Passive Avoidance Test

On days 50 and 51, all animals were tested for the STPA task [56,57]. The apparatus is a two-compartment (15 × 20 × 15 cm high) box, where one is illuminated with white polyvinylchloride walls, and the other is darkened with black polyvinylchloride walls and a grid floor. A guillotine door separates each compartment. A 60 W lamp positioned 40 cm above the apparatus lights up the white compartment during the experiment. Scrambled foot shocks (0.3 mA for 3 s) were delivered to the grid floor using a shock generator scrambler (Lafayette Instruments, Lafayette, IN, USA). The guillotine door was initially closed during the training session. During the training session, each mouse was placed into the white compartment. After 5 s, the door was raised. When the mouse entered the dark compartment and placed all its paws on the grid floor, the door was closed, and the foot shock was delivered every 3 s. The step-through latency, that is, the latency spent to enter into the dark compartment, and the number of vocalizations were recorded. The retention test was carried out 24 h after training. Each mouse was placed again into the white compartment. After 5 s, the door was raised. The step-through and escape latencies (corresponding to the re-exit from the dark compartment) were recorded up to 300 s. Animals that show latencies during the training and retention session, both lower than 10 s, are considered as failing to respond to the procedure and are discarded from the calculations. In this study, no animal was discarded accordingly.

### 4.6. Biochemical Analysis

On day 51, all animals were sacrificed by decapitation without prior anesthesia after the behavioral session. For all animals, trunk blood was collected into EDTA-coated tubes that were inverted several times and placed on ice until plasma separation through centrifugation (2000 g for 10 min at 4 °C). The brain was quickly removed, and the hippocampus and frontal cortex were dissected out on an ice-cold metal plate. On n = 6 animals per group, the hippocampus was used to determine the lipid peroxidation levels (LPO), and the cortex was used for measuring TNFα and IL-6. The other brain structures and plasma were frozen on dry ice and stored at −80 °C at Amylgen for two months and available for supplementary biochemical assays.

Regarding LPO analysis, the procedure was modified and adapted from Hermes-Lima et al. (1995) [58]. This method measures the ability of brain-peroxidized lipids to oxidize a ferrous oxide/xylenol orange complex. This procedure was named FOX assay (ferrous oxidation/xylenol orange method) and was shown to be a fast and accurate method to measure lipid peroxides. A comparison between the FOX method and the TBARS (thiobarbituric acid reactive substances) assay was made previously by Hermes-Lima et al. (1995). The authors determined that the TBARS assay detects products at different stages of lipid peroxidation, whereas the FOX assay has the advantage of direct detection of lipid peroxides. Six (6) hippocampi from each group were used. After thawing on ice, samples were homogenized by ultrasound (1 kJ, 10 s, Sonopuls, Bandelin, France) in cold methanol (1/10 *w*/*v*), centrifuged at 1000× *g* for 5 min, and the supernatant was placed in Eppendorf tubes. The reaction volume of each homogenate was added to FeSO_4_ 1 mM, H_2_SO_4_ 0.25 M, xylenol orange 1 mM and incubated for 30 min at room temperature. After reading the absorbance at 580 nm (A5801), 10 µL of cumene hydroperoxide (CHP) 1 mM was added to the sample and incubated for 30 min at room temperature to determine the maximal oxidation level. The absorbance was measured at 580 nm (A5802). The level of lipid peroxidation is determined as CHP equivalents according to CHPE = A5801/A5802 × [CHP (nmol)] and expressed as CHP equivalents per wet weight of tissue and as a percentage of the mean BC group data.

Regarding brain and plasma inflammatory markers levels, contents in IL6 and TNFα were analyzed by ELISA assays. For all assays, the cortex was homogenized after thawing in 50 mM Tris-150 mM NaCl buffer, pH 7.5, and sonicated for 20 s. After centrifugation (16,100× *g* for 15 min, 4 °C), supernatant or plasma was used for ELISA assays according to the manufacturer’s instructions. For each assay, absorbance was read at 450 nm, and sample concentration was calculated using the standard curve. Results are expressed in pg of marker per mg of tissue. Then, the results were expressed as a percentage of the mean BC group data. N = 10 per group were assayed.

### 4.7. Statistical Analysis

All values, except passive avoidance latencies, were expressed as mean ± S.D. (Standard Deviation). Statistical analyses were performed separately for each compound using one-way ANOVA (F value), followed by Tukey’s post hoc multiple comparison test. Passive avoidance latencies do not follow a Gaussian distribution since upper cut-off times are set. They were therefore analyzed using a Kruskal-Wallis non-parametric ANOVA (H value), followed by a Tukey’s multiple comparison test. Values with *p* < 0.05 were considered statistically significant. For Morris Water Maze data analysis, a Tukey’s multiple comparison test was performed after two-way ANOVA.

## 5. Conclusions

In conclusion, the present results underscore the positive impact of the standardized extract of *Phaeodactylum tricornutum* containing 2% of fucoxanthin on cognitive function parameters, including spatial working memory and short-term memory, achieved through the regulation of oxidative stress and inflammation pathways. To better understand the underlying mechanisms of actions regulated by the microalgae PT extract, several studies are needed focusing on mitochondria function, Nrf2 pathway activity, microglial cells of the central nervous system, and the gut-brain axis. Overall, these data are promising, advocating for the promotion of the microalgae extract of PT as a natural and viable alternative in the dietary supplement area for improving cognitive function in humans.

## Figures and Tables

**Figure 1 marinedrugs-22-00099-f001:**
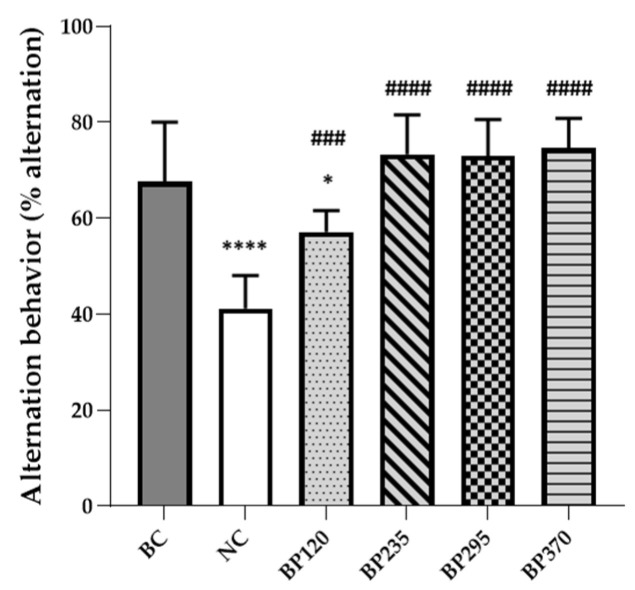
Effects of microalgae extract of PT on D-Gal-induced spontaneous alternation deficits in mice. *n* is between 11 and 12 depending on the groups; * *p* < 0.05, **** *p* < 0.0001 vs. the Saline solution/Normal food group (BC group), ### *p* < 0.001, #### *p* < 0.0001 vs. the D-GAL 150/Veh group (NC group); One-way ANOVA (F value), followed by Tukey’s post hoc multiple comparison test. BC: Blank control group; NC: Negative control group; BP120: dose of 120 mg/day/70 kg of BrainPhyt^TM^; BP235: dose of 235 mg/day/70 kg of BrainPhyt^TM^; BP295: dose of 295 mg/day/70 kg of BrainPhyt^TM^; BP370: dose of 370 mg/day/70 kg of BrainPhyt^TM^.

**Figure 2 marinedrugs-22-00099-f002:**
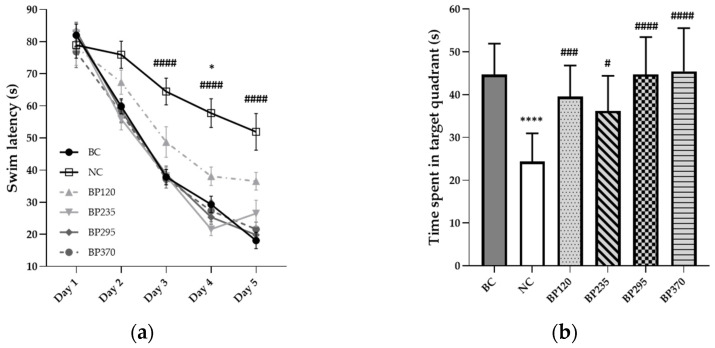
Effects of microalgae extract of PT on D-Gal-induced learning deficits in mice. (**a**) swin latency (seconds). (**b**) time spent in the target quadrant (seconds). n is between 11 and 12 depending on the groups; * *p* < 0.05 vs. the Saline solution/Normal Food group (BC group); **** *p* < 0.0001 vs. the Saline solution/Normal food group (BC group), # *p* < 0.05, ### *p* < 0.001, #### *p* < 0.0001 vs. the D-GAL 150/Veh group (NC group);Tukey’s multiple comparison test after two-way ANOVA. BC: Blank control group; NC: Negative control group; BP120: dose of 120 mg/day/70 kg of BrainPhyt^TM^; BP235: dose of 235 mg/day/70 kg of BrainPhyt^TM^; BP295: dose of 295 mg/day/70 kg of BrainPhyt^TM^; BP370: dose of 370 mg/day/70 kg of BrainPhyt^TM^.

**Figure 3 marinedrugs-22-00099-f003:**
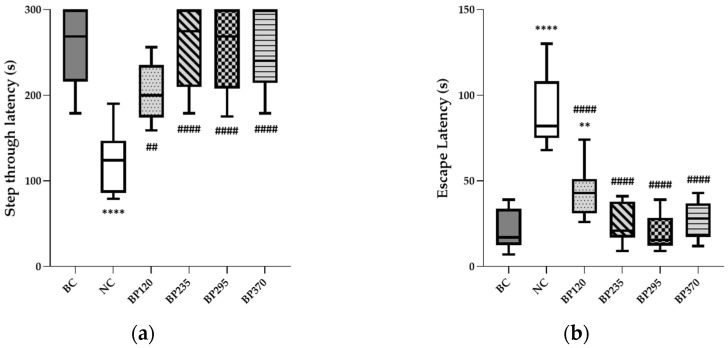
Effects of Microalgae extract of PT on D-Gal-induced contextual long-term memory deficits in mice. (**a**) Step through latency (seconds). (**b**) Escape latency (seconds). n is between 11 and 12 depending on the groups; ** *p* < 0.01, **** *p* < 0.0001 vs. the Saline solution/Normal Food group (BC group), ## *p* < 0.01, #### *p* < 0.0001 vs. the D-GAL 150/Veh group (NC group); One-way ANOVA (F value), followed by Tukey’s post hoc multiple comparison test. BC: Blank control group; NC: Negative control group; BP120: dose of 120 mg/day/70 kg of BrainPhyt^TM^; BP235: dose of 235 mg/day/70 kg of BrainPhyt^TM^; BP295: dose of 295 mg/day/70 kg of BrainPhyt^TM^; BP370: dose of 370 mg/day/70 kg of BrainPhyt^TM^.

**Figure 4 marinedrugs-22-00099-f004:**
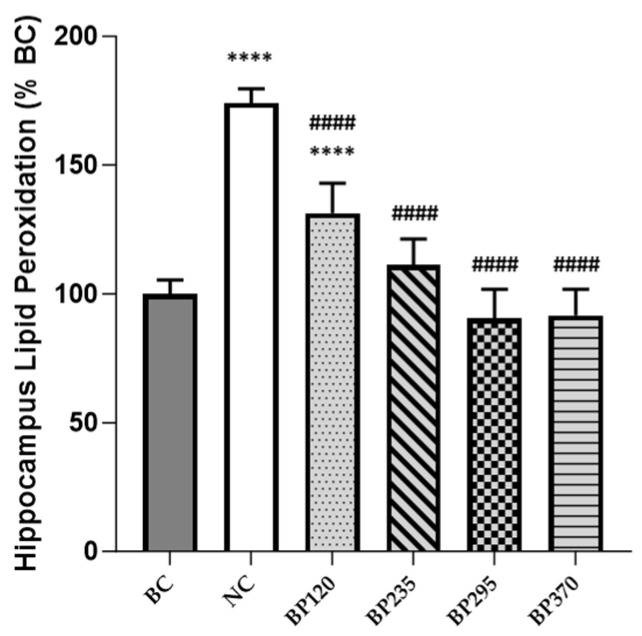
Effects of microalgae extract of PT on D-Gal-induced lipid peroxidation increase in mice hippocampus. n is 6 per group; **** *p* < 0.0001 vs. the Saline solution/Normal Food group (BC group), #### *p* < 0.001 vs. the D-GAL 150/Veh group (NC group); One-way ANOVA (F value), followed by Tukey’s post hoc multiple comparison test. BC: Blank control group; NC: Negative control group; BP120: dose of 120 mg/day/70 kg of BrainPhyt^TM^; BP235: dose of 235 mg/day/70 kg of BrainPhyt^TM^; BP295: dose of 295 mg/day/70 kg of BrainPhyt^TM^; BP370: dose of 370 mg/day/70 kg of BrainPhyt^TM^.

**Figure 5 marinedrugs-22-00099-f005:**
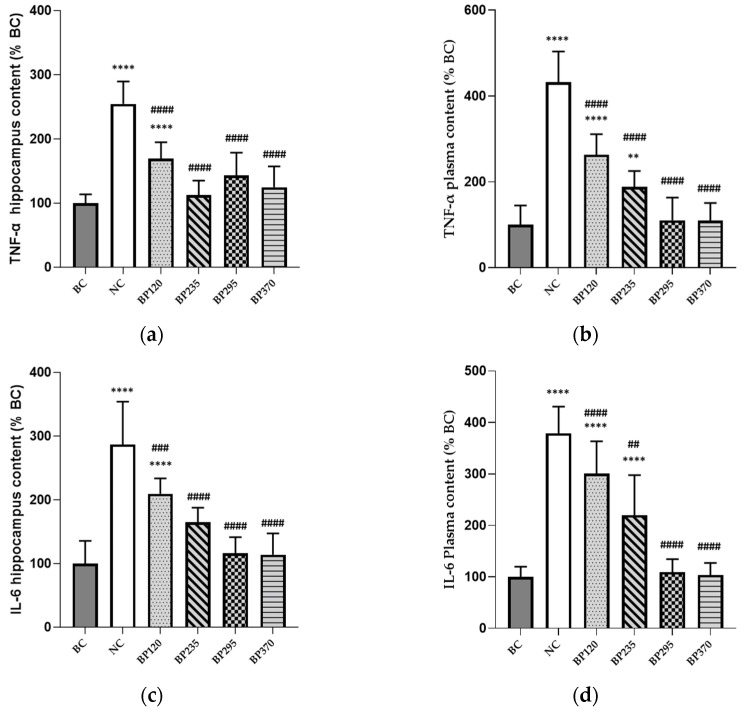
Effects of microalgae extract of PT on D-Gal-induced brain and plasma inflammatory markers increase in mice. (**a**) TNF-α brain content (% BC). (**b**) TNF-α plasma content (% BC). (**c**) IL-6 brain content (% BC). (**d**) IL-6 plasma content (% BC). n is 10 per group; ** *p* < 0.01, **** *p* < 0.0001 vs. the Saline solution/Normal Food group (BC group), ## *p* < 0.01, ### *p* < 0.001, #### *p* < 0.0001 vs. the D-GAL 150/Veh group (NC group); One-way ANOVA (F value), followed by Tukey’s post hoc multiple comparison test. BC: Blank control group; NC: Negative control group; BP120: dose of 120 mg/day/70 kg of BrainPhyt^TM^; BP235: dose of 235 mg/day/70 kg of BrainPhyt^TM^; BP295: dose of 295 mg/day/70 kg of BrainPhyt^TM^; BP370: dose of 370 mg/day/70 kg of BrainPhyt^TM^.

**Figure 6 marinedrugs-22-00099-f006:**
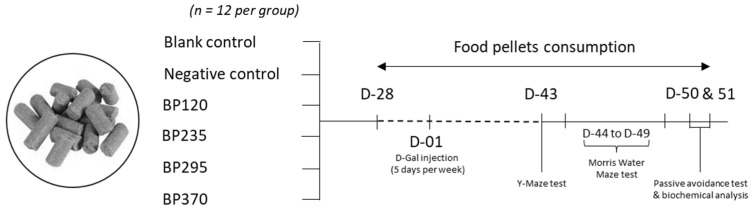
Study experimental design. Mice were randomly divided into blank control group (BC), negative control group (NC), and 4 microalgae extract of PT (branded name: BrainPhyt^TM^) groups with different human equivalent doses evaluated (BP120, dose of 120 mg/day/70 kg; BP235, dose of 235 mg/day/70 kg; BP295, dose of 295 mg/day/70 kg; BP370, dose of 370 mg/day/70 kg) with 12 mice in each group.

**Figure 7 marinedrugs-22-00099-f007:**
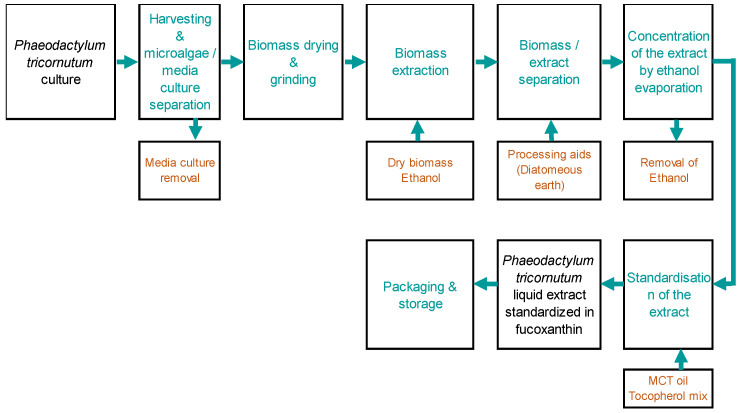
Microalgae extract of PT production process by Microphyt company.

**Table 1 marinedrugs-22-00099-t001:** Microalgae extract of PT global specifications.

	Composition (%)
*Phaeodactylum tricornutum* extract	40–70 (w:w)
MCT oil based on coconut oil	30–60 (w:w)
Mix on non-GMO tocopherols	0.45–0.55 (w:w)
Total lipids	60–90 (w:w)
Proteins	5–15 (w:w)
Humidity	<2 (w:w)
Ashes	<10 (w:w)
Carbohydrates	0.5–20 (w:w)
Total PUFAs-w3	≥4 (w:w)
All-trans-fucoxanthin	2.0 ± 0.4% (w:w)

**Table 2 marinedrugs-22-00099-t002:** Main chlorophylls, total carotenoids, and fatty acids composition of microalgae extract of PT. Data are presented as mean ± SD from six independent industrial-scale batches.

	Concentration
**Chlorophylls***-* g equivalent chloropyll A/100 g)	
Chlorophyll *c*	3.46 ± 0.45
Σ Chlorophyll *b*	ND
Σ Chlorophyll *a*	0.24 ± 0.25

**Total carotenoids**	
*Β-carotene - g equivalent Β-carotene/100 g*	0.09 ± 0.08
All trans Fucoxanthin *-* g equivalent fucoxanthin/100 g	1.99 ± 1.01

**Total unsaturated fatty acids omega 3**—g/100 g of product	6.51 ± 1.17
Eicosapentaenoic acid (EPA)	7.99 ± 6.14
Docosahexaenoic acid (DHA)	0.23 ± 0.20
Alpha-linolenic acid (ALA)	0.24 ± 0.17
Stearidonic acid (SDA)	0.07 ± 0.08

**Total unsaturated fatty acids omega 6**—g/100 g of product	2.56 ± 0.19
Hexadecadienoic acid	2.01 ± 1.38
Linoleic acid	0.83 ± 0.66
Arachidonic acid	0.52 ± 0.44
Eicosatetraenoic acid	0.14 ± 0.12

**Total saturated fatty acids**—g/100 g of product	45.42 ± 8.49
Caprylic acid	23.24 ± 5.35
Capric acid	18.23 ± 4.06
Palmitic acid	2.53 ± 0.60
Myristic acid	0.90 ± 0.25

## Data Availability

Raw data are available from the corresponding author upon justified requests.

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
