# Peer review of "A Standardized Extract of Microalgae Phaeodactylum tricornutum (Mi136) Inhibit D-Gal Induced Cognitive Dysfunction in Mice"

_marinedrugs, 2024, doi:10.3390/md22030099_

Round 1
Reviewer 1 Report
Comments and Suggestions for Authors
The paper entitled ‘A standardized extract of Microalgae Phaeodactylum tricornutum (Mi136) inhibit D-Gal induced cognitive dysfunction in mice’ presents interesting studies results based on mice model and biochemical assay. The issue is well presented. On the whole, the paper is well prepared, concise and informative. Authors decided to use three in vivo assay which are adequate for aim of the studies. In my opinion the paper is suitable for publication in the Journal with minor modifications:
Abstract: informative and concise, including the most significant information
Introduction: well prepared, informative
Results: presented in form of figures and detail description of obtained results along with statistical analysis. All figures include full description of signs.
Discussion: detail, based on up-to-date references. Authors discussed obtained results in a substantive and factual way. Please provide information wheter the similar extracts were studied with other in vivo model. Please provide more information about studies based on rats.
Methods: 1) whether the mice were selected for sex? 2) why authors decided to use D-Gal? There are a few substances which can be used in similar studies.
Conclusions: lack. Please provide
Author Response
Thank you for your comments and positive feedbacks. To the best of our knowledge, similar extracts from Phaeodactylum tricornutum were not studied with other in vivo model. Regarding other microalgae and as mentioned in the discussion section, the effects of Spirulina and Astaxanthin (from haematococcus pluvialis) extracts on brain health were evaluated. For example, many in vivo studies on different animal models using spirulina have highlighted neuroprotective effects in different brain areas (Sorrenti and al., 2021 – Mar. Drugs). We added thes references in our manuscript. However, we think that it’s difficult to compare the results obtained in our study with other considering the methodology differences (e.g. supplementation duration, different behavior tests, aging mice model used, biomarkers evaluated etc.). Thus, for this reason, we decided to not specifically develop this part in our discussion section to focus on potential mechanisms of actions regulated by the microalgae extract evaluated in the present study.
Sorrenti, V.; Castagna, D.A.; Fortinguerra, S.; Buriani, A.; Scapagnini, G.; Willcox, D.C. Spirulina Microalgae and Brain Health: A Scoping Review of Experimental and Clinical Evidence. Mar Drugs 2021, 19, 293, doi:10.3390/md19060293.
Liu, H.; Zhang, X.; Xiao, J.; Song, M.; Cao, Y.; Xiao, H.; Liu, X. Astaxanthin Attenuates D-Galactose-Induced Brain Aging in Rats by Ameliorating Oxidative Stress, Mitochondrial Dysfunction, and Regulating Metabolic Markers. Food Funct. 2020, 11, 4103–4113, doi:10.1039/D0FO00633E.
Regarding your question on mice sex, we used 72 male swiss mice as mentioned in the Methods section. In fact, as the large majority of in vivo studies, we used only male mice primarily to avoid the physiological variability linked with the estrous cycle of female rodents in particular.
We agree with you that there are few other interesting substances as Lipopolysaccharide-Induced Neuroinflammation (LPS) which could be used as accelerated model of aging in mice. Based on literature analyzis other subastances like β-amyloid peptide seem to mimic more specifically pathological models such as Alzheimer's. Thus, we decided to use D-Gal for the following reasons :
-Induce both oxidative stress and inflammation through different signalling targeted by the present microalgae extract and as largely discussed in the manuscript. For example, LPS induces a major immune response and focus specifically on the production of pro-inflammatory cytokines
-As mentioned in the review from Cai and al. (2022, published in Cells), D-Gal induces specific brain aging by increasing mitochondrial dysfunction, oxidative stress and inflammation which mimics various behavioral and molecular features occuring during aging (age-related disorders). While other substances as LPS were more used for age-related diseases based on our literature analyzis.
-Induces oxidative stress and brain disorders over a prolonged period. This is totally in line with the study's objective which was to evaluate the preventive effect of microalgae extract.
Finally as you rightly suggested, we added a specific conclusions section at the end of the manuscript.
We hope that our answers will meet your expectations.
Reviewer 2 Report
Comments and Suggestions for Authors
Algae currently represent a significant point of consideration for scientists worldwide. On the one hand, they are a rich source of food, while on the other hand, their cultivation (while maintaining marine water purity) is highly efficient and relatively uncomplicated on an industrial scale. Additionally, algae have proven to be an incredibly valuable source of nutrients and even active substances used in current pharmacology. Hence, this project is an interesting piece discussing the impact of algae as antioxidants on reducing the rate of neurodegeneration. However, there are noticeable inconsistencies, which I hope to clarify to improve the overall perception of the work.
The main difficulty in following the text of the publication comes with the overwhelming number of abbreviations (BP, NC, PT), and furthermore, doses marked in numbers 120-370 which are not explained. It is also unclear why specific doses (amounts) of extract were used. Continuing with the aspect of abbreviations, every abbreviation used in the figure must be explained in its description. The reader should be able to decipher every piece of information contained in the figure.
My biggest concern is the limited assessment of the pathway by which algae's influence on galactose neurotoxicity. Behavioral tests postulate neurodegeneration and disturbances in memory processes unequivocally linked to neuronal activity. Then suddenly, there is an examination of the global induction of oxidative stress caused by galactose, although the source (i.e., the type of cells stimulated) is omitted. And ultimately, we end up with the secretion of pro-inflammatory cytokines IL-6 and TNF-alpha. Overall, it is unclear why, out of all the possibilities, these particular cytokines were chosen. Additionally, they are labeled in the brain (astroglial and microglial production) and circulating blood (macrophage and lymphocyte production). The cellular sources of TNF-alpha are once again inexplicably combined. After all, TNF-alpha in circulating blood has extremely limited effects in the brain due to the existence of the blood-brain barrier. The authors do not point to a specific scientifically sanctioned reason, almost as if they were saying, "We chose these cytokines because everyone studies them in Alzheimer's disease."
In fact, galactose induces mitochondrial dysfunction and disturbances in cellular OXPHOS. This leads to neuronal death and the replenishment of astrocytes and microglia. The high efficacy of algae in behavioral tests suggests the absolute necessity of examining each cell group, which is clearly lacking in the project. Constructive conclusions cannot be drawn from the publication beyond one - algae have antioxidant effects, which has been known for a very long time. The publication shows chaos in the design and interpretation of the study's results. Hence, the overall scientific value of the entire project is low.
Author Response
Thank you for your reviewing and comments. We totally agree that there is a growing interest to find ecological solutions and to meet major societal challenge as the acceleration of the general population ageing and the use of microalgae as sustainable molecule of interest sources is a recent promising approach. Please find below some answers :
1/ Regarding abbreviations and as you requested, we add the following description for each figures « BC: Blank control group ; NC: Negative control group ; BP120: dose of 120 mg/day/70kg of BrainPhytTM; BP235: dose of 235 mg/day/70kg of BrainPhytTM; BP295: dose of 295 mg/day/70kg of BrainPhytTM; BP370: dose of 370 mg/day/70kg of BrainPhytTM. ». As we are aware of the large number of abbreviations used, we have recalled their meaning regularly throughout the manuscript, whether for the terms Phaeodactylum tricornutum (PT) or Fucoxanthin (FX).
Regarding your comments on study doses, we determined 4 different experimental doses to evaluate potential dose-response effect and support the determination of an optimize dose for human study perspectives. And as you can on our results we found a dose-response with higher benefits in in more elevated doses. But it’s very interesting to note that even with the lower doses, we showed significant benefits in all behavior and biochemical parameters. Also, we defined the doses in a relatively low range of supplementation (human equivalent) to meet the doses authorized for human consumption.
2/ The main research hypothezis was that a supplementation with a microalgae extract from Phaeodactylum tricornutm limit cognitive impairment induced by D-Galactose, a well recognized accelerated aging model. This hypothezis was based (as precised in the introduction section) on several previous works suggesting a neuroprotective role of fucoxanthin in particular, the main molecule of interest in the ingredient evaluated in the present study. But to our knowledge, no previous studies have investigated precisely the impact of a microalgae extract from Phaeodactylum tricornutum on cognitive function parameters. Thus, before to evaluate the effects on several specific mechanism of actions as you justly mentionned in your comments, we wanted to proof the efficacy of this extract on behavior tests and overall oxidative stress and inflammatory well recognized to be involved in neurodegeneration process and altered following D-Galactose. Also, we totally agree with you that TNF-alpha and IL-6 in circulating blood has extremely limited effects in the brain due to the existence of the blood-brain barrier but we used this blood parameter to « validate » the use of D-Galactose as accelerated aging model in mice.
As you may have read in the Discussion section of the present manuscript, we totally aware about the impact of galactose on mitochondria function, astrocytes and microglia and we obviously precised that further studies are needed to better understand the precise mechanisms of actions underlying the regulation of inflammatory and oxidative stress pathway through the microalgae extract from Phaeodactylum tricornutum. Thus, for us it’s a very interesting mechanistic study but it’s another and complementary works.
Regrading the evaluation of brain biomarkers, we decided to specifically target hippocampus as major brain region contributing to alterations in learning, spatial working and episodic memories. Based on the interesting review published by Shwe and al. (2018, Experimental Gerontology) the main inflammatory markers used for monitoring in D-galactose-induced aging models are cyclooxygenase (COX-2), iNOS, NOS-2, tumor necrosis factor alpha (TNF-α), interleukin (IL-1β), IL-6, nuclear factor (NF-κB) thioredoxin-interacting protein (Txnip), p-NF-κBp65, p-IκBα, p-IKKα, p-IKKβ. In the same way, this literature review identified a non-exhaustive list of oxidative stress biomarkers (e.g. MDA, 8-OxoG, ROS, NOS, Protein carbonyl etc.). And we decided to evaluate lipid peroxidation, TNF-α and IL-6 based on the main results showed in previous studies evaluating the cognitive function and related findings on oxidative stress and inflammation parameters in D-galactose-induced aging models.
To clearly highlight the conclusions that can be drawn from the present publication, we added the following paragraph at the end of the manuscript (section 5. Conclusions).
« In conclusion, the present results underscore the positive impact of the standardized extract of PT containing 2% FX on cognitive function parameters, including spatial working memory and short-term memory, achieved through the regulation of oxidative stress and inflammation pathways. To better understand the underlying mechanisms of actions regulated by the microalgae PT extract, several studies are needed fo-cusing on mitochondria function, Nrf2 pathway activity, microglial cells of the central nervous system and also the gut-brain axis. Overall, these data are promising, advocat-ing for the promotion of the microalgae extract of PT as a natural and viable alterna-tive in the dietary supplement area for improving cognitive function in humans.»
We hope that these answers will improve your perception of the objectives of this work. We totally understand and share your expectations to better understand the underlying mechanism of actions and in the coming months we will be conducting research project specifically dedicated to this objective.
Reviewer 3 Report
Comments and Suggestions for Authors
The manuscript by Jonathan Maury et al., entitled “A standardized extract of Microalgae Phaeodactylum tricornutum (Mi136) inhibit D-Gal induced cognitive dysfunction in mice” reported that the microalgae Phaeodactylum tricornutum (PT) containing 2% FX on cognitive function parameters such as spatial working memory, long-term memory, and short-term memory through the regulation of oxidative stress and inflammation pathways. The manuscript is well-written and the cited references are appropriate. The manuscript needs minor revision before its final publication.
Comments:
1. The authors should include the clear schematic diagram for the experimental design.
2. In each Figure, biochemical analysis authors should mention the graph ‘x’ axis legend as “specific brain region (eg., hippocampus or cortex)” instead of “Brain content”.
3. The authors should analyze the cognitive related markers expression in the brain samples through western blot method to support the effect of Microalgae Phaeodactylum tricornutum treatment.
4. The authors should include the reference for the behavior tests.
5. Minor editing is needed throughout the manuscript.
Author Response
Thank you for your relevant comments and positive feedbacks. Please find below our answers.
1/ We added a specific figure to show the experimental design (Figure 6 in section 4.1).
2/ In each figure related to biochemical analyzis, we replace « Brain content » by « Hippocampus content »
3/ We agree that the use Western blots method (semi-quantitative) would be interesting to support the results shown on brain and blood biomarkers evaluated. Regarding lipid peroxidations analyzis, we decided to use FOX assay method (and not TBARS method) as justified in the section 4.6. Regarding cytokines markers, we decided to use ELISA assays (specific quantitative method) as we targeted specific proteins (TNF-α and IL-6), we had low sample volumes. Also ELISA method are highly specific and sensitive assays used to detect concentrations of as little as 0.01 nanograms of antigen or antibody per milliliter of sample. Thus, we are confident on the robustness of the results presented in this study regarding biomarkers even if we agree that in next studies it could be very interesting to use the 2 methods (quantitative and semi-quantitative).
4/ We included some references for the behavior tests in the Material and Methods sections
5/ Thank you for your reviewing
We hope that our answers will meet your expectations.
Round 2
Reviewer 2 Report
Comments and Suggestions for Authors
No further comments.
Author Response
Thank you for your time and reviewing.